# Understanding the Relationship of the Human Bacteriome with COVID-19 Severity and Recovery

**DOI:** 10.3390/cells12091213

**Published:** 2023-04-22

**Authors:** Hassan Zafar, Milton H. Saier

**Affiliations:** 1Department of Molecular Biology, School of Biological Sciences, University of California, San Diego, CA 92093-0116, USA; msaier@ucsd.edu; 2Central European Institute of Technology, Masaryk University, 625 00 Brno, Czech Republic

**Keywords:** SARS-CoV-2, COVID-19, bacteriome, immune system

## Abstract

The Severe Acute Respiratory Syndrome Coronavirus-2 (SARS-CoV-2) first emerged in 2019 in China and has resulted in millions of human morbidities and mortalities across the globe. Evidence has been provided that this novel virus originated in animals, mutated, and made the cross-species jump to humans. At the time of this communication, the Coronavirus disease (COVID-19) may be on its way to an endemic form; however, the threat of the virus is more for susceptible (older and immunocompromised) people. The human body has millions of bacterial cells that influence health and disease. As a consequence, the bacteriomes in the human body substantially influence human health and disease. The bacteriomes in the body and the immune system seem to be in constant association during bacterial and viral infections. In this review, we identify various bacterial spp. In major bacteriomes (oral, nasal, lung, and gut) of the body in healthy humans and compare them with dysbiotic bacteriomes of COVID-19 patients. We try to identify key bacterial spp. That have a positive effect on the functionality of the immune system and human health. These select bacterial spp. Could be used as potential probiotics to counter or prevent COVID-19 infections. In addition, we try to identify key metabolites produced by probiotic bacterial spp. That could have potential anti-viral effects against SARS-CoV-2. These metabolites could be subject to future therapeutic trials to determine their anti-viral efficacies.

## 1. Introduction

The severe acute respiratory syndrome coronavirus-2 (SARS-CoV-2) causes a corona virus disease (COVID-19), which first emerged in Wuhan, China in 2019 [1]. As of March 2023, the virus had caused more than 600 million morbidities with over 6 million mortalities around the globe [2,3], information about the COVID-19 pandemic is presented in Figure 1. A member of the Coronaviridae family, this RNA virus is the seventh Coronavirus with the ability to cause infections in humans [4]. The other six include severe acute respiratory syndrome coronavirus-1 (SARS-CoV-1), and middle eastern respiratory syndrome (MERS-CoV)—both of which cause serious infections in humans, while the remaining four, human coronavirus HKUI (HcoV-HKU1), human coronavirus NL63 (HcoV-NL63), human coronavirus OC43 (HcoV-OC43) and human coronavirus 229E (HcoV-229E), are associated with mild infections [5]. The COVID-19 pandemic has caused a huge threat to global public health and has caused economic losses throughout the world [6].

COVID-19 in humans may be associated with asymptomatic infections or mild respiratory symptoms [7,8]. However, in some instances, it may progress to severe pneumonia, which increases the chances of mortality. The biological mechanisms behind the mild and severe disease forms are still not well-understood and are currently subject to further research. However, old age and other co-morbidities appear to be predisposing factors for acquiring severe pneumonia associated with COVID-19 [9]. The major issue when coping with SARS-CoV-2 is its rapid dissemination, which occurs mainly through the spread of oral droplets [10,11]. The virus has increased dissemination potential in crowded environments, where human to human interactions are at their maximum.

Currently, the COVID-19 pandemic could be on its way to an endemic form. However, this assumption may not be accurate as the endemicity of a virus depends on various factors, including the demographics, population susceptibility, immune status of the people, and emergence of new viral variants. As explained by Cohen and Pulliam, in the long run, most COVID-19 infections may occur in people that were either previously infected (thus, having a stronger protection) or/and vaccinated. This presumed pattern of infection would result in a lower number of hospitalizations (as has been observed with past coronaviruses) and potential mortalities [12].

The human bacteriome (sum of all bacterial spp. residing in the human body) has been subject to extensive research over the past two decades. The availability of recent genome mining tools such as “Metagenomics and Metatranscriptomics” (see Box 1), has increased our understanding of the physiology, metabolism, and interactions of the microbial residents in humans. The bacteriome affects human health both positively (beneficial or probiotic microbes) and negatively (pathogenic microbes) [13]. The well-known beneficial roles include modulation of the immune system, maintenance of organismal homeostasis, host nutritional assistance, and antagonism of pathogenic microbes [14]. On the negative side, a dysbiotic bacteriome may cause pathogenesis by assuming a role of secondary invaders of, for example, the intestinal epithelia. Published studies have indicated a positive correlation between respiratory viral infections and the microbial composition of the lungs and gut [15,16]. Similarly, a correlation may exist between COVID-19 and the composition of the human microbiota. Other studies have suggested that in systemic infections, SARS-CoV-2 has the potential to infect enterocytes in the intestines and cause diarrhea [17]. In addition, a recent report points to a relationship between a disrupted gut bacteriome and COVID-19 severity [18]. 

Box 1Metagenomic and Metatranscriptomic tools to study the human bacteriome.Studies on the human bacteriome have evolved over the past few decades [19]. Various molecular and bioinformatic tools can now be used to study the bacterial communities inhabiting the human body. Two of these routinely used methods are “metagenomics”, and “metatranscriptomics”.
*Metagenomics*
Metagenomic tools enable biologists to study the entire genetic material in a bacterial community [20]. After extraction of microbial DNA, Next Generation Sequencing (NGS) is used, which produces huge datasets in the form of short reads. Analyzing the data is analogous to putting pieces of a puzzle together, which enables the acquisition of information about the taxonomic profile of the bacterial community [21]. Various computational tools to study metagenomic data include QIIME and MOTHUR [22,23].
*Metatranscriptomics*
This deals with the analysis of the transcriptome of bacterial species in a natural environment [24]. Metatranscriptomic tools help to elucidate the functional potential of bacteria and identify metabolic pathways of importance at the host–microbe interface. It is now possible to perform whole metatranscriptomic shotgun sequencing, the expression and functional profiling of a microbiome [25]. Metatranscriptomic reads are generally mapped to specialized databases such as KEGG, and Uni-PROT [26,27].

The human immune system has two types of immune responses (innate and adaptive) to an external microbe that enters the human body [28]. The immune system also counters toxic substances that may have entered the human body through mucosal surfaces. In addition to the mobilization of the protectors (immune cells) of the human body against various microbial invaders, the immune system also helps to distinguish between self- and non-self-components such as cells, proteins, and sugars [17]. In Figure 2, we have introduced important immune cells that are pivotal to the immune response against SARS-CoV-2. For more information about the role of immune cells in human health, we advise the readers to review past literature [18,29].

Mucosal immunity is localized and has a specific organization. It provides protection to the inner surfaces of the body. It spans various organ systems including the gastrointestinal tract (GIT) and the respiratory tract, to name a few [34]. However, according to the anatomical location in the body (oral cavity, nasal cavity, lungs, and gut), the immune cells of the mucosal immune system may differ in types and mechanisms of activation [35]. In addition, there is a plethora of information about how the bacterial residents of the human body and viruses can modulate the immune system in a negative or positive manner [36,37,38]. 

In this review, we shall consider potential relationships between altered oral, nasal, lung, and gut bacteriomes with COVID-19 severity. Reports concerning the association of COVID-19 with dysbiosis in human bacteriomes will be considered. We shall try to arrive at a consensus regarding healthy bacteriomes and identify pathogenic bacterial spp. that may contribute to disease severity. Additionally, potential probiotic bacteria and their respective metabolites will be identified that may be helpful in promoting recovery from COVID-19. Through this communication, we shall also try to clarify the positive and negative roles of bacterial diversity in different human body locations during the progression of this disease. Lastly, we will try to identify the key roles the immune system plays during COVID-19 infections, and how the immune system and bacteriome may team-up as a pair to either fight SARS-CoV-2 or help in its systemic dissemination in the human body. 

## 2. Is the Oral-Cavity a Reservoir for SARS-CoV-2?

The infectivity of SARS-CoV-2 is mainly dependent on entry of the virus into the human body, which depends on the presence of angiotensin converting enzyme-2 (ACE2) and Transmembrane Serine Protease-2 (TMPRSS2) receptors on target cell surfaces [39]. The structure of SARS-CoV-2 and its interaction with the ACE2 receptor is shown in Figure 3, while the TMPRSS2 is currently subject to structural studies. Interestingly, these two receptors are scattered throughout the human body in various organs (heart, bladder, kidney, nasal cavity, etc.) and are not restricted to the respiratory system [40,41]. With respect to the oral cavity (one of the primary entry points of the virus), both receptors have been identified in salivary glands (SG) and the oral mucosa [42]. Both SG and the oral mucosa are shown in Figure 4. Noteworthy is the finding that the expression of ACE2 receptors is higher in the SG (parotid, submandibular, and sublingual (the major SG), labial, buccal, glossopalatine, palatine and lingual (the minor SG)) as compared to the lungs. These observations may indicate that these glandular locations are predilection sites for viral persistence [43].

The oral mucosa consists of three portions: (i) the masticatory; (ii) the lining; and (iii) the specialized, and these regions contain the ACE2 receptor for the binding of the SARS-CoV-2 spike protein [47]. Past studies suggested the prevalence of mucosal lesions in COVID-19 patients [48]. The higher membrane fusion activity of the virus in the oral cavity, due to increased expression of ACE2 sheddases; a disintegrin and metalloprotease 17 (ADAM17) and a disintegrin and metalloprotease 10 (ADAM10), and endopeptidases; Calpain-1 catalytic subunit (CAPN1), and Calpain small subunit 1 (CAPNS1) suggests that the oral cavity is a reservoir for the virus [49]. Published data on viruses indicate that the gingival sulcus may be a preferred residency site for numerous viruses including herpes simplex virus, Epstein Barr virus and the human cytomegalovirus [50]. As this site is also a preferred ecological niche for numerous bacterial residents, it can be assumed that a symbiotic association is in play here. In agreement with this suggestion, Gupta et al. found that the gingival crevicular fluid harbors SARS-CoV-2 virions [51]. 

It has been hypothesized that the periodontal pocket is a preferred site of localization for active and latent forms of SARS-CoV-2. The virus may replicate in the periodontium, reach the oral cavity and saliva, and spread hematogenously via the periodontal capillaries to distant organs in the body [52]. This suggests that conditions such as periodontitis may lead to recurrent systemic infections in COVID-19 patients. The results of Matuck and co-workers indicated the persistence of SARS-CoV-2 in the periodontal tissues of patients who had long courses of the disease [53]. Although the sample size (n = 9) of this post-mortem study was low, this observation could reflect a long-term persistence of the virus in periodontal tissues of patients. Similar results were observed by To et al. who also noticed viral particles in the posterior oropharyngeal samples of patients 20 days post-infection [54]. Overall, the oral cavity can be regarded as an underestimated reservoir for SARS-CoV-2 infection and transmission. 

In a recent study, the role of the oral cavity in virus spread and transmission was explored by Huang et al. [55]. The authors used single cell RNA sequencing and in situ hybridization to develop two datasets for the minor SG and gingiva (9 samples, 13,284 cells, and 50 cell clusters). They observed that the epithelial cells of the SG and gingiva contain large numbers of the viral receptors, ACE2 and TMPRSS2. Similar observations have been reported by Matuck et al., who observed viral SARS-CoV-2 particles in the ductal lining, cell cytoplasm, acinar cells, and ductal lumen of the SG [56]. It can safely be assumed that the SG includes underexplored sites for the spread and replication of the virus. SARS-CoV-2 could propagate in the SG, thereby sustaining the virus in other anatomical sites. This might not be of surprise, as the SG has also been associated with the pathogenesis of other viruses, including Ebola virus (EBV), Human Herpes virus 7 (HHV-7), and cytomegalovirus (CMV) [57,58,59]. Viral replication within the SG seems to assist in dissemination of the virus, as contaminated droplets expelled during coughing, sneezing, and speech are rich in salivary excretions [60]. 

## 3. The Oral Bacteriome and COVID-19: What Do We Know?

The human oral bacteriome is the second largest bacterial community in the human body, after that of the gut, and it includes around 700 recognized species [61,62]. The oral bacteriome has been regarded as an important player in the establishment of infection caused by viruses that enter the body via the oropharynx [63]. In the oral cavity, respiratory viruses encounter bacterial residents and are modulated in their ability to establish infection [64]. Moreover, viruses can alter the balance of the oral microbiota, thus promoting dysbiosis. A dysbiotic oral bacteriome is often associated with periodontal inflammation, which could lead to local and systemic disease conditions, including those sustained by viral infections [65].

As noted, before, numerous bacterial spp. make up the oral bacteriome; thus, the oral cavity may be regarded as an ecological community of bacterial commensals, symbionts, and potential pathogens [66]. The primary bacterial genera residing in the oral bacteriome include *Capnocytophaga*, *Corynebacterium*, *Fusobacterium*, *Leptotrichia*, *Neisseria*, *Prevotella*, *Streptococcus*, and *Veillonella* [44]. Oral bacteria have been related to respiratory infections in several ways: (i) oral pathogens can be aspirated into the lungs, (ii) enzymes secreted by periodontal pathogens can modify mucosal surfaces, resulting in increased colonization and adhesion by respiratory pathogens, and (iii) cytokines secreted in response to periodontal pathogens can alter respiratory epithelia, thus promoting the colonization of pathogens [67,68]. 

Metagenomic analysis of the oral bacteriome of patients suffering from COVID-19 have revealed the abundance of cariogenic (tooth decay) and periodontopathic (periodontitis) bacteria [69]. This indicates that changes in the diversity of the oral bacteriome can lead to COVID-19 complications. Periodontopathic bacteria have been associated with respiratory infections and other chronic inflammatory pathologies including diabetes, hypertension, and cardiovascular diseases [45]. These diseases have also been reported to exhibit co-morbidities associated with complications and mortalities due to COVID-19. The well-known putative periodontopathic organisms include *Actinobacillus actinomycetemcomitans*, *Eikenella corrodens, Bacteroides forsythus, Bacteroides gingivalis, Bacteroides intermedius,* and *Wolinella recta* [70,71]. Analyses of the oral bacteriomes of COVID-19 patients can give information about indicator species that increase in number during infections. If these species have pathogenic potential, they may cause complications associated with the disease. 

A study by Ward and co-workers identified indicator species of the oral bacteriome as predictors of COVID-19 severity in patients [46]. Three bacterial spp. that appear to be associated with disease severity include *Porphyromonas endodontalis, Veillonella tobetsuensis,* and *Bifidobacterium breve.* However, as this research was based on disease modeling, further clinical research is imperative in order to confirm the association of these bacterial species with disease progression. Interestingly, *P. endodontalis* was observed to be the most important discriminator of COVID-19 severity. Conversely, the abundance of *Muribaculum intestinale* in patients was indicative of more moderate COVID-19 infections. In a recent study by Miller et al., minimal differences were observed between the oral bacteriomes of newly admitted COVID-19 patients and non-COVID-19 patients [72]. Sequencing of the 16S rRNA gene was performed to make a comparative analyses of the bacterial diversity between the positive and the control patients. The authors observed increased abundance of *Prevotella pallens* in the positive patients, while *Rothia mucilaginosa*, and *Streptococcus* spp. were abundant in the control patients. Species abundant in the saliva samples of the control patients included *Prevotella denticola*, *Prevotella oris*, *Saccharibacteria* strain HMT356, and *Streptococcus peroris.* The high and low viral loads in the saliva of patients corresponded to the distribution of the different bacterial spp. These included *Prevotella pallens*, *Stomatobaculum* spp., *Streptococcus infantis*, *Streptococcus parasanguinis* clade 411, *Streptococcus sanguinis*, and *Treponema* spp. The authors also hypothesized that the relationship between the bacteriome and viral saliva load in the patients could be affected by the receipt of supplemental oxygen. 

A comparison of the oral bacteriomes of healthy people and COVID-19 patients along with immunological analyses of their cytokine levels could provide valuable information about beneficial bacterial residents of the oral cavity, allowing discrimination of cytokine levels raised during SARS-CoV-2 infection. In this regard, a study by Iebba et al. analyzed 16S rRNA sequencing for samples taken from the oral cavity [73]. These investigators identified various bacterial species as potential biomarkers for COVID-19 severity. These included *Prevotella jejuni*, *Prevotella salivae*, *Soonwooa purpurea*, and *Veillonella infantium*. Bacterial spp. that predominated the oral bacteriome were *Gemella taiwanensis*, *Granulicatella elegans*, *Kallipyga gabonensis*, *Neisseria perflava*, *Porphyromonas pasteri*, *Rothia mucilaginosa*, and *Streptococcus oralis*. Results of the in silico analyses also predicted the above aforementioned bacterial spp. as probiotics, which could assist in controlling COVID-19 severity and the associated cytokine storm. The authors also identified six COVID-19-related discriminant cytokines including Interleukins (IL)-2, 5, 6, Granulocyte colony-stimulating factor (GCSF), Tumor necrosis factor-α (TNF- α), and Interferon-γ (IFN- γ), but only IL-12 for controls. Both IL-6 and -12 were the most discriminant cytokines for positive and control patients. Past literature suggested that during viral immune responses, IL-6 can be overexpressed, thus leading to impaired functionality of T-helper cells [74]. Due to this constant antigen stimulation (as in the case of SARS-CoV-2), cluster of differentiation 8 (CD8) T-cells do not respond to the antigenic stimuli as they normally would and as a consequence memory CD8 T-cells do not form—a situation that limits viral clearance [75]. As reported by Iebba et al., the predicted probiotic bacterial spp. had a negative correlation with IL-6. This could indicate that these organisms may help lower the pro-inflammatory environment of the oral cavity, and potentially help in countering the cytokine storm associated with COVID-19. 

Another recent metagenomic analysis identified enrichments in opportunistic oral pathogens, *Megasphaera* and *Veillonella* in COVID-19 patients. However, no significant changes in alpha-diversity were observed during the comparison of the non-critically ill patients to healthy controls [76]. Findings from various studies highlight the necessity for further comprehensive studies on the oral bacteriome in COVID-19 patients. It is also pivotal to establish a clear picture of how a dysbiotic oral bacteriome may be an important player in disease severity. It is the need of the hour to identify key probiotic strains that may aid in the recovery from this disease. 

Studies on the relationships between respiratory viruses and the oral bacteriome indicated that the virus–bacterium interaction could enhance disease severity [77]. For instance, the interaction between neuraminidase-producing streptococci and influenza virus has been shown to increase the viral load [78]. Similar trans-kingdom interactions are expected for SARS-CoV-2 and the oral bacteriome of humans, as similar interactions have been reported in other bacteriomes of the human body [72]. Martino and co-workers analyzed the oral bacteriome of COVID-19 patients and observed changes in the normal bacterial communities in comparison to controls [79]. There was an abundance of bacterial species capable of modifying heparan sulfate, a component that is essential for the binding of SARS-CoV-2 to ACE2. Similar interactions have been reported for the gut bacteriome (discussed in later sections), as gut residents may influence the synthesis of various cofactors that are needed for viral binding to ACE2. In addition, inflammation-causing bacterial species (*Streptococcus mutans* and *Prevotella nigrescens)* have been associated with a dysbiotic oral bacteriome [80]. It is possible that COVID-19-mediated inflammation may cause a change in the normal oral bacteriome and increase the number of pathogens, which may cause further inflammation. At this point, it is imperative to further analyze the interplay of the immune system with the normal and/or dysbiotic oral bacteriome during COVID-19 infection.

Past studies showed that oral hygiene improves the symptoms of patients suffering from pneumonia while reducing the mortality rate. Sjogren et al. suggested that good oral hygiene could prevent one in ten deaths of older patients (65 years and older) suffering from pneumonia [81]. Mori et al. observed that hygienic oral practices can prevent the incidence of ventilator-associated pneumonia in intensive care units [82]. As of now, the search continues for the establishment of a defined relationship between the oral bacteriome and COVID-19 severity. However, oral dysbiosis could be a modifiable risk factor for COVID-19, as hygienic oral practices may circumvent dysbiosis. It seems that these practices should be adopted for public health promotion during the COVID-19 pandemic and even during future COVID-19 endemics. Our current understanding of the relation between the dynamic trio (COVID-19, oral bacteriome, and the immune system) based on literature discussed in this section is summed up in Figure 4 and Figure 5.

## 4. The Nasal Bacteriome Is Still Underexplored!

The nasal cavity (NC) is part of the upper respiratory tract and consists of three divisions: (i) vestibule, (ii) respiratory, and (iii) olfactory [83]. An anatomical scheme is provided in Figure 6. There are curved shells of bones projecting from the lateral walls of the nasal cavity called “turbinates”. These turbinates create four pathways for the flow of air into the nasal pathway in the (i) inferior meatus, (ii) middle meatus, (iii) superior meatus, and (iv) spheno-ethmoidal recess [83]. In this section of the review, we shall focus on regions of the nasal cavity that have been reported to harbor bacterial communities and try to draw conclusions from data about the relationships between nasal bacteriomes and COVID-19 severity.

The NC provides an important pathway to and from the external environment [87]. Through this pathway, microbes enter the human body daily. The NC is home to various microbes including commensals, symbionts, and pathogens [88]. Various factors such as temperature and humidity may help explain the diverse population of microbes in the NC. The bacteriome of the anterior nares is abundant in three phyla: Actinobacteria, Firmicutes, and Proteobacteria [89]. At the genus level, members of *Corynebacterium*, *Moraxella*, *Propionibacterium*, and *Staphylococcus* are dominant. However, the middle meatus is dominated by three species: *Propionibacterium acnes, Staphylococcus aureus,* and *Staphylococcus epidermidis* [90]. Previous reports on alterations of the nasal bacteriome in response to viral infections suggest changes in the bacterial ecosystem of the NC, and an increase in pathogenic bacterial spp. [91]. 

Chronic rhinosinusitis is caused by infection with rhinovirus [92]. Lal et al. compared the bacteriomes of the middle and inferior meatus in chronic rhinosinusitis patients (with and without nasal polyps) with controls [93]. They found the nasal samples of patients with chronic rhinosinusitis (without nasal polyps) were enriched with the genera *Fusobacterium*, *Haemophilus*, and *Streptococcus,* and they exhibited less bacterial diversity compared to the controls. The samples collected from the middle meatus of chronic rhinosinusitis patients with nasal polyps were dominated by three genera (*Alloiococcus, Corynebacterium,* and *Staphylococcus*) along with a decrease in bacterial diversity. Other studies on the relation of rhinovirus infection and the nasal bacteriome suggested similar relationships (a decrease in bacterial diversity during viral infection) [93,94]. 

Overall, comprehensive data are lacking about changes in the diversity of the nasal bacteriome during viral infections. Also of importance is the anti-viral immune state and how this state increases the chances of bacterial infection of the upper respiratory tract.

## 5. COVID-19 and the Nasal Bacteriome

As discussed previously, the composition of the nasal bacteriome heavily influences the progression of viral respiratory tract infections. However, detailed information about the potential role that the nasal bacteriome might play during COVID-19 progression is lacking. Nevertheless, it is now clear that the nasal barriers are among the first lines of defense against SARS-CoV-2 [95]. The commensal microbes of the NC can help limit the emergence and spread of opportunistic pathogens by selective inhibition, and by producing metabolites for niche establishment [96]. Past literature indicates that various mechanisms can be used by viruses to control bacterial spp. and their metabolites for dealing with any potential threats in their surroundings. As a result, the virus can cross the host–cell barriers in their dissemination to different anatomical locations in the body [97]. In essence, any virus that enters the human body can affect the bacterial residents in the body, and these bacteria could either control or be disrupted by the invading virus. This in turn could lead to either viral suppression or stimulation [98].

In a study by Nardelli et al., a comparison of the nasopharyngeal samples of COVID-19 patients and healthy controls was conducted [99]. These investigators observed reduced populations of Proteobacteria and Fusobacteria in COVID-19 patients with higher abundances of *Fusobacterium periodonticum* in the control group. Similarly, Moore and co-workers found a decrease in populations of *F. periodonticum* in nasal samples of COVID-19 patients 10 days post-hospitalization [100]. *F. periodonticum* has been reported to play a potential role in the surface sialylation process [101]. It has been suggested that sialic acid residues of this bacterium could function as alternative receptors for the S-protein of SARS-CoV-2, whose preferred receptor for binding is ACE2 [84]. This points to the notion that a protective mechanism is mediated by the bacterial sialome against viral infections, where reduced sialic acid hydrolysis from glycoproteins and glycolipids could lead to a decrease in protection against COVID-19. As other species of *Fusobacterium* have been shown to exhibit strong adherence to human cells, in turn modulating host immune/inflammatory responses, this could be a reason behind a negative correlation between the abundance of *F. periodonticum* and COVID-19 disease severity [99]. As of now, this hypothesis remains to be tested, to establish the role of *F. periodonticum* as a bacterium of interest in the fight against COVID-19. 

According to the literature currently available, there seems to exist a negative correlation between microbial diversity within the NC and the occurrence of severe COVID-19 infections [102]. Smith et al. reported a similar relation in their analyses, which included 16S rRNA sequencing of the nasal microbiota [85]. However, in the critically ill patients, there was also a decline in the populations of beneficial commensals such as *Corynebacterium* and *Dolosigranulum* as compared to controls. Additionally, the critically ill patients had an abundance of pathogenic genera including *Staphylococcus*, *Prevotella* and *Peptostreptococcus.* These results agree with Gupta et al., who showed decreased microbial diversity in COVID-19 patients [103]. In addition, an abundance of opportunistic pathogens such as *Acinetobacter*, *Haemophilus*, *Moraxella*, *Pseudomonas* and *Stenotrophomonas* was observed. 

It is important to integrate the nasal bacteriome with the abundances of nasal cytokines to obtain a clear picture of the complexity of interactions between host, virus and commensals that could be associated with disease severity. In this respect, Smith et al. integrated 16S rRNA results of the nasal bacteriome with spike protein specific immune responses (cytokines and antibodies) in COVID-19 patients and controls [85]. They found that cytokines that decreased (IL-33, IFN-λ3 and IFN-γ) or increased (epidermal growth factor) with SARS-CoV-2 infection were linked to overall microbial α-diversity and to the presence of *Corynebacterium,* suggesting genus-specific and community-driven regulation of mucosal cytokine production. Previously, it had been observed that the expression of interferons by plasmacytoid dendritic cells could be modulated by a normal bacteriome. This suggests that a normal or beneficial bacteriome could help alleviate COVID-19 symptoms [101]. Simultaneously, the predominance of pathogens such as *Staphylococcus* in the NC had a strong correlation with IL-6 mediated systemic inflammation [86,104]. It may be assumed that dysbiosis in the nasal bacteriome of patients may be a pre-disposing factor for systemic inflammation in COVID-19. Such patients may be subject to severe COVID-19 infections as epithelial barriers may be compromised, and the pathobionts of the dysbiotic nasal bacteriome may disseminate to other body locations, thus causing secondary infections. In light of recent literature, a scheme of our understanding of the relationships between the nasal bacteriome, the immune system, and SARS-CoV-2 is depicted in Figure 6. 

## 6. Relationship of the Lung Bacteriome with the Immune System

Initially, the lungs were considered as sterile organs, and this dogma persisted in the scientific community for many decades [105]. However, advancements in microbiological tools not only proved that the lungs are not sterile, but that they harbor a diverse bacterial community [106]. The physiological process of respiration involves rounds of air inhalation and exhalation. As a result of these processes, numerous microbes enter and leave the lungs daily. In general, bacteria first enter the body through the upper respiratory tract and then gain access to the lungs by direct mucosal dispersion and micro-aspiration. There is a constant migration of microbes from the upper to the lower respiratory tract (the lungs) [89]. Accordingly, there are many factors of mucosal immunity at play with the bacterial inhabitants of the lungs. It has been suggested that the bacterial diversity in the lungs of healthy individuals is somewhat similar [107]. However, in comparison to the gut bacteriome (proven to have important roles in immune responses and modulation), there is a paucity of information on how bacterial spp. regulate the immune cells of the lungs. 

Human lungs harbor low numbers of microbes, approximately 2.2 × 10^3^ bacterial genomes per cm^3^ [108]. The maintenance of a small diverse bacteriome seems to be pivotal for health and prevention of pulmonary diseases [109]. Numerous factors such as environmental conditions, antibiotic therapies, health status, genetics, smoking, and preferred breathing behavior (nose or mouth) may affect the overall bacterial picture of the lungs [102]. In past studies, it has been shown that the overgrowth of a single bacterial species leads to a decrease in overall bacterial diversity [110]. The microbial imbalance of the lung bacteriome has been associated with the progression of various diseases such as cystic fibrosis [111,112]. In a nutshell, it seems as if a balanced, stable lung bacteriome is important for the prevention of pulmonary diseases. 

According to the literature, the major phyla abundant in the lungs are Bacteroidetes, Firmicutes, Proteobacteria, Fusobacteria and Actinobacteria [113,114]. In young children, the dominant species in the lung bacteriome are of the genera *Hemophilus*, *Moraxella*, *Staphylococcus* and *Streptococcus* [115]. In healthy adults, the species *Prevotella*, *Streptococcus*, and *Veillonella* dominate the bacterial landscape of lungs [116]. However, due to variations in physiological parameters (oxygen tension, pH, temperature, etc.) of the lung environment, the growth and selection of the normal bacterial residents, mostly commensals, could be affected. This in turn could lead to a lack of spatial bacterial diversity, allowing a specific bacterial sp. to outgrow other microbes in the altered lung environment [117]. To date, there is a lack of understanding about the role of a healthy lung bacteriome on healthy respiration; however, there seem to be intricate relationships at play between the mucosal immunity of the lungs and the bacteriome. 

The immune cells of the lungs normally keep the pathogens in the airways of the lungs in check [118]. However, their major immunological role is to circumvent the overproduction of inflammatory responses to harmless environmental stimuli. A primary characteristic of the lung microenvironment is high immune tolerance, which is mainly controlled by alveolar macrophages and dendritic cells [119]. The immunoregulation by these cells results in the generation of regulatory T-cells as well as the release of IL-10, prostaglandin E2, and tumor growth factor-β [120]. There is increasing evidence that the lung bacteriome directly influences the immune response and contributes to immune tolerance [121]. Pathogen pattern receptors recognize ligands of both commensal and pathogenic bacteria. As a result of recognition, two types of signals are developed: (i) danger signals and (ii) safety signals. The first of these two type of signals is in response to pathogens and calls on pro-inflammatory cytokines, while the latter is in response to commensal bacteria or non-damaged self-tissues [122]. The discrimination between pathogenic and commensal and/or beneficial bacteria by the immune system is of obvious importance. The commensal microbes can sense the increase of pathogens in their environment and antagonize the spread of the invaders, keeping their ecological niches intact [123]. The commensals also have another way to contribute to immune tolerance in the lungs as they do not fully penetrate the lung mucus layer and are therefore excluded from the epithelial lining [124]. As a result, the commensal residents are not recognized by pathogen pattern receptors. However, the pathogens can easily penetrate the mucus layer and disseminate in the epithelium. They achieve this by using virulence factors that are at their disposal, and as a result, they can be recognized by pathogen pattern receptors on immune cells, thus resulting in pro-inflammatory responses [122]. 

Studies using murine models have shown an increase in the bacterial population in lungs during the first two weeks of COVID-19 [124,125]. Additionally, there is a transition from the normal resident phyla (Gammaproteobacteria and Firmicutes) to Bacteroidetes [126]. Such changes in the bacteriome are related to the accumulation of a programmed death ligand -1- dependent T-regulatory cell population, that may promote immune tolerance during antigen and/or allergen challenge [127]. Results from murine studies show a close association between the lung bacteriome and immune tolerance [121,123,128]. Additionally, the acquisition of a healthy lung bacteriome is a pivotal life event, which has a long-term effect on human health, as it helps to protect the lungs from foreign antigens and pathogens. This also holds true for the lung bacteriome of human neonates [129]. In a study with adults, Segal and co-workers reported a strong correlation between bacterial spp. (*Prevotella* and *Veillonella*) in the lungs and elevated levels of lymphocytes [130]. They also observed higher levels of inflammation mediated by T-helper-17 cells, and a lower Toll like receptor (TLR) response from alveolar macrophages. In addition, it has been shown that in patients with acute respiratory distress syndrome there is a positive correlation between the abundance of the phyla Proteobacteria and inflammation (both alveolar and systemic) [131,132]. 

## 7. COVID-19 and the Lung Bacteriome

In comparison to other bacteriomes within the human body, the anatomical site of sampling is pivotal with respect to the lung bacteriome, as direct sampling of the alveolar space (primary site of disease) is an arduous task. The preferred two methods of sampling the lungs directly are (i) tracheal aspiration and (ii) bronchoscopy [133] (shown in Figure 7). In the former procedure, a tracheal aspirate is collected by passing a plastic catheter into the trachea, which induces coughing of the patients, and another catheter is passed for collection of the secretions. However, the complications associated with tracheal aspiration include bleeding and emphysema [134]. In bronchoscopy, a tube (bronchoscope) is passed down the mouth and larynx into the alveolar space, which is sampled via bronchoalveolar lavage [135]. However, this method of sampling could be risky owing to the aerosol properties of the virus, and contamination by the sampling personnel constitutes an unfortunate possibility. In this section, we focus only on recent studies that used either tracheal aspiration or bronchoscopy for sample collection from COVID-19 patients as these two methods should give a better estimate of the dynamics of the lung bacteriome in patients.

In a recent report, Hernandez-Teran and co-workers used tracheal aspiration to analyze the lung bacteriome in different groups of people (healthy and non-COVID-19 pneumonia as well as mild, severe, and fatal COVID-19 patients) [138]. In the severe and fatal COVID-19 groups, they observed an increase in the abundance of anaerobic genera such as *Abiotrophia, Mycoplasma,* and *Streptococcus*. As a result of a viral respiratory infection, the inflammatory process increases mucus production, which favors biofilm production and the growth of anerobic genera. This could be a reason why these bacteria were abundant in severe and fatal COVID-19 groups. In other groups, interesting bacterial abundances were observed; for instance, in the healthy group, the normal diversity of the lung bacteriome was maintained with the genera *Oribacterium, Streptococcus*, and *Veillonella* (the most abundant genera). In the non-COVID-19 pneumonia group, the genus *Corynebacterium*, a nosocomial pathogen, was most abundant. In the mild group, bacteremia causing spp. such as *Prevotella melaninogenica*, *Veillonella parvula* and *Neisseria subfava* were represented with increased numbers. In the fatal COVID-19 group, the significant bacterial species were *Rothia dentocariosa*, *Streptococcus infantis*, and *Veillonella dispar*. Additionally, in the severe COVID-19 group, there were increased populations of the genus *Megasphera*. Although *Streptococcus* is a commensal resident of the lung bacteriome, it may cause pathogenesis during environmental disturbances. Species from the genus *Rothia* are residents of the human oral bacteriome; however, species of this genus have also been identified as opportunistic pathogens [139]. According to previous reports, the genus *Megasphera* has been associated with ventilator-associated pneumonia [140]. This information is of substantial interest, and future studies will enhance our understanding of the possibility that increased populations of *Megasphera* in the lungs contribute to COVID-19 severity. 

Metagenomics and metatranscriptomics are useful tools in the estimation of functional potentials (DNA-based) and functional activities (RNA-based) of a bacteriome. In a recent study, Suleiman et al. used bronchoscopy for the collection of bronchoalveolar lavage of COVID-19 patients and analyzed bacterial species in the samples using metagenomic and transcriptomic tools [136]. The patients were split into three groups based on clinical outcomes; these were (i) survivors with ≤28 days on mechanical ventilation, (ii) survivors with >28 days on mechanical ventilation, and (iii) deceased patients. In addition, they analyzed the viral load and profiled the immune response to SARS-CoV-2. They observed a distinct composition of the lung bacteriome as compared to samples from the upper respiratory tract and controls. In the meta-transcriptome data, the abundant bacterial spp. were *Finegoldia magna, Staphylococcus epidermis*, *Staphylococcus aureus, Mycoplasma salivarium, Bacillus thuringiensis, Prevotella oris,* and *Streptococcus anginosus*, while the metagenome datasets showed increased abundance of *Staphylococcus aureus, Salmonella enterica, Burkholderia dolosa, Klebsiella variicola, Xanthomonas citri,* and *Aeromonas hydrophila*.

Interestingly, *Mycoplasma salivarium*, a normal oral commensal, was abundant in the deceased patients and those on mechanical ventilation for >28 days. The spread of this organism from the oral cavity to the lungs could be a consequence of micro-aspiration. This specie is of high interest as it may control immune cells of the lungs and may play a role in increasing the pathogenesis of SARS-CoV-2. Previous studies on *Mycoplasma* spp. have suggested an increase in their abundance in patients with ventilator-associated pneumonia [141], although the causative agents of this condition are *Staphylococcus pneumonia* and *Pseudomonas aeruginosa* [142,143]. In a more detailed study, Nolan et al. observed increased abundances of *M. salivarium* in the bronchoalveolar lavage of patients with ventilator-associated pneumonia [141]. Macrophages phagocytose microbes and other particulate matter, and they are a key part of pulmonary defenses [144]. Immunological assays with *M. salivarium* showed that it significantly impaired phagocytosis by macrophages [141]. By characterizing the host response in the bronchoalveolar lavage specimens, Sulaiman and co-workers showed that alveolar concentrations of anti-spike and anti-receptor-binding domain antibodies were decreased in deceased patients, which indicates a negative correlation with viral replication in the lungs [136]. In addition, they observed down-regulation of immunoglobulin (IgA and IgG) production. Interestingly, the findings of Sulaiman et al. conflict with the widespread belief that lung injury during COVID-19 is due to an uncontrolled cytokine storm. Thus, the study sets in motion the idea that a virus-specific immune deficiency may contribute to non-resolving lung injury during COVID-19. Statistically significant differences were noted only in the meta-transcriptomic data and not in the meta-genomic data. These observations suggest that functional activation of microbes can provide further insights into the environment of the lung bacteriome of patients with fatal outcomes. In the meta-transcriptomic data, the major differentially expressed bacterial pathways in the poor outcome groups involved glycosylases, oxidoreductases, transporters, and two-component sensor kinase-response regulatory systems (signaling systems used extensively by bacteria) [136]. Additionally, in the deceased group, there was increased expression of antibiotic resistant genes among lung bacteria as compared to the other groups. This indicates that pathogenic and/or antibiotic resistant bacterial spp. could be at the forefront of secondary infections in the lungs.

In another study, Tsitsiklis et al. collected lung samples via tracheal aspiration and analyzed the dynamics of the lung bacteriome by RNA-seq [137]. They observed that individual immune responses to SARS-CoV-2 infection restructured the bacterial communities of the lung bacteriome, increasing susceptibility to ventilator-associated pneumonia. Patients having ventilator-associated pneumonia showed increased IFN-1 production and dysregulated antibacterial immune signaling. In addition, a decrease in the activities of macrophages, neutrophils, and T-cells was observed. Additionally, decreased TLR signaling led to impaired activation of key cytokines (IL-1, 6, 8, 17) for defense against pathogens. This state of immune suppression disrupts the bacteriome of the lungs, resulting in the overgrowth of pathobionts. 

The diversity and stability of the lung bacteriome appears to be important for normal host–bacterium interactions [116]. The eubiosis of the lung bacteriome is important for host health and homeostasis [145]. For instance, the literature suggests a homeostatic mechanism in the lung epithelia that could assist in a “IFN primacy state” in the lungs. This antiviral state in the lungs could help prevent attacks from respiratory viruses such as influenza [146]. Members of the bacteriome can induce pathogen-associated molecular patterns that result in the antiviral state [138]. Additionally, in other viral infections, various factors and changes (alterations in the epithelia, increased binding ability of viral pathogen, and bacterial dysbiosis) can promote disease severity [138], and the same could be the case for SARS-CoV-2. Our current understanding, based on the literature presented in this review about the interplay of the lung bacteriome, the immune system, and COVID-19 is summarized in Figure 7. 

## 8. The Human Gut Bacteriome

The human gut harbors most of the microbial residents in the human body [147,148]. It is sometimes considered by scientists to be a separate fully functional organ in humans, which contributes to many important physiological processes such as absorption, metabolism, susceptibility, and resistance to various kinds of diseases, xenobiotic responses, and immunomodulation [149]. The gut bacteriome (GB) starts to develop at birth; however, the bacterial composition and diversity in neonates is determined by numerous factors such as mode of birth (vaginal or cesarian), milk feeding habits (breast or formula milk), and transitions to semi-solid or solid diets [149,150]. The adult GB is mostly governed by dietary-habits, and there seem to be variations in the bacterial genera present according to geographical locations and types of diets [151,152]. Over the past two decades, the GB has been extensively studied using the latest bioinformatic tools, see [148,149,150,151,153]. In the gut, the bacterial residents can live as symbionts (commensals and mutualists), and in certain environmental conditions, as pathogens [154]. Additionally, the locations of the microbes are highly important; for instance, a bacterium may be a symbiont in the gut, but translocation to extra-abdominal locations may result in pathogenesis and subsequent harmful effects to the host [155]. Over 50 phyla have been identified in the GB, the two major phyla being *Bacteroidetes* and *Firmicutes*, while minor phyla include *Proteobacteria*, *Fusobacteria*, *Tenericutes*, *Actinobacteria* and *Verrucomicrobia* [156]. However, the densities of microbial populations differ not only in various parts of the gut (small and large intestines, caecum, and the colon and rectum of the large intestine), but also in the epithelial linings and lumen of the intestine [157,158].

The major genera present in the stomach are *Lactobacillus, Veillonella* and *Helicobacter* [159]. However, the small intestine harbors a complex microbial community, having less bacterial diversity and abundance (≈10^3^–10^7^ bacterial cells/gram) in comparison to the much denser bacterial population in the colon (162 and see below). This is due to numerous environmental challenges faced by microbes passing through the small intestine such as a low pH (pH ~2–5), higher concentrations of oxygen, influxes of bile, the presence of variable concentrations of antimicrobial peptides, and secretory immunoglobulins, secreted from the small intestinal epithelia [160]. Due to the harsh conditions in the proximal intestine, the bacterial populations that thrive there are less diverse, have a lower biomass and are highly dynamic [161]. The bacterial communities of the small intestine are rapidly growing facultative anaerobes that have the metabolic repertoire to tolerate the combined effects of bile acids and antimicrobial peptides [162]. Bile acids, such as antimicrobial peptides, can be bactericidal to certain species due to their surfactant properties. They are considered to be important players in shaping the bacterial landscape of the small intestine [160]. Despite facing harsh environmental conditions, bacterial residents of the small intestine compete effectively for simple carbohydrates that are available in their respective niches [163]. Meta-transcriptomic analyses have shown that various metabolic processes in the small intestine are highly active in comparison to those in the large intestine [164]. This results in a rapid fluctuation of nutrient availability in the lumen where simple carbohydrates are rapidly metabolized for bacterial community maintenance. This contrasts with the colon, where bacterial residents tend to degrade and utilize complex carbohydrates [165]. 

The large intestine (particularly the colon) has the densest bacterial populations in the GIT [166]. The bacterial population in the colon is about 10^12^ cells/gram [167]. Some reasons for the enhanced colonization of the large intestine by microbes include less host-mediated nutrient adsorption, a fairly neutral pH (pH ~6–7), and sub micromolar levels of oxygen [168]. The large intestine is mainly inhabited by anaerobes, which are adept in the degradation and utilization of polysaccharides, glycoproteins, and glycolipids (complex carbohydrates) [165]. The distal parts of the GIT are dominated by the phylum Bacteroidetes, which have sophisticated metabolic machineries for the degradation of carbohydrates (See review [169]). A scheme of the human gut with dominant bacterial genera is depicted in Figure 8.

## 9. The Human Gut Bacteriome and COVID-19

Among all the bacteriomes of the human body, the relation of the GB with COVID-19 has been most extensively studied. In 2020, Lamers and co-workers showed that SARS-CoV-2 could infect enterocytes in the intestine, as these cells readily express the receptor ACE2 [170]. Most of the studies have involved (i) sequencing of stool samples from COVID-19 patients and comparing them with those from healthy controls, (ii) metagenomic and meta-transcriptomic analyses, and (iii) in vivo animal models by infecting non-human primates with SARS-CoV-2. 

In recent metagenomic and metatranscriptomic studies, Sun and coworkers analyzed fecal samples from sixty-three COVID-19 patients and eight healthy controls [171]. In addition, immunohematological parameters were analyzed to provide insight into inflammation biomarkers and the status of immune cells. In COVID-19 patients, there was an increase in the phylum Verrucomicrobia, while members of the dominant phylum, Firmicutes, had decreased populations. With regard to proven beneficial bacteria, in the COVID-19 group there was a decrease in the species *Alistipes shahii, Bacteroides cellulosilyticus, Bacteroides eggerthii, Bifidobacterium pseudocatenulatum, Eubacterium eligens Faecalibacterium prausnitzii*, and *Lawsonibacter asaccharolyticus.* These species help in maintaining a physical barrier between the microbe and the host and help in preventing the dissemination of foreign pathogens through anti-microbial peptides and secretion system-dependent bacterial antagonism. Additionally, there were high numbers of opportunistic pathogens including *Acinetobacter bereziniae, Bacteroides ovatus*, and *Clostridium innocuum* in the COVID-19 patients. In a comparison between the COVID-19 patients having severe versus mild symptoms, disparate patterns of microbial abundance were observed. In the group with severe systemic infections, there were increased abundances of *Bacteroides nordii*, *Bifidobacterium longum, Blautia sp.* CAG 257, and *Burkholderia contaminans*. According to the meta-transcriptomic data, the dominant metabolic pathways in some of these bacteria included glycolysis, sugar fermentation, and the biosynthesis of methionine, vitamin B_12_, and teichoic acid. The enrichment of glycolytic pathways among the gut bacteriome could be a reason for increased inflammation in COVID-19 patients, as these pathways had previously been associated with SARS-CoV-2 infection [172]. During various microbial infections in the human body, there is increased activation of macrophages and dendritic cells, while the preferred metabolic pathway changes from using lipids to using sugars [173]. The generation of ATP via glycolysis is important due to the engagement of the TLR with the related activation of the phosphoinositide-3-kinase-protein kinase-B pathway [174]. These same authors observed upregulated microbial virulence genes that could assist in motility, adherence, and escape of the microbes from immune responses. These bacterial pathogenicity factors have the potential to translocate through the leaky gut into the circulatory system, promoting the secretion of inflammatory cytokines by activating pattern recognition receptors (TLRs and Nucleotide oligomerization domain-like receptors), thus leading to systemic inflammation. Additionally, different correlations between the microbial spp. and immune biomarkers were observed. In this regard, the strongest associations for the immune markers were found for the two opportunistic pathogens *Burkholderia contaminans* and *Bacteroides nordii.* For instance, there was a negative association between *Burkholderia contaminans* and T-cell populations, T-cell responses and complement protein C3. On the other hand, there was a positive correlation of the pathogen with increased blood levels of the C-reactive protein. This protein is produced due to extensive inflammation of blood vessels and is a general marker for measuring inflammation in the body. It is routinely measured for COVID-19 patients [175]. In the case of *B*. *nordii*, there were positive correlations with all of the white blood cells, in particular neutrophils. 

It has been established that COVID-19 is a multi-organ disease with a wide array of clinical manifestations. Even after patient recovery from the virus, there can be prolonged and persistent health effects. The term coined for this post-COVID-19 condition is, “Post-Acute COVID-19 Syndrome” (PACS) (see Box 2). To date, not many studies have been conducted on the status of the bacteriomes in patients that underwent PACS after recovery from SARS-CoV-2. However, a recent study did shed light on the gut bacteriome status of recovered patients having PACS [176]. The authors analyzed the stool samples of 106 COVID-19 patients at admission, after one month and after six months, and compared the status of the gut bacteriome with COVID-19 patients who did not undergo PACS, and with healthy controls. Interestingly, they found that the profile of the gut bacteriome at admission governs the susceptibility of the patients to suffer long-term complications of COVID-19. For instance, at admission, the patients who did not undergo PACS had a diverse bacterial composition of the gut with the prevalent genera including *Bacteroides, Bifidobacterium*, and *Blautia*. On the other hand, at admission, the gut bacteriome of the patients that underwent PACS was less diverse and had reduced abundances of numerous beneficial bacteria including *Bifidobacterium longum,* and *Blautia wexlerae* along with high numbers of *Actinomyces johnsonii*, *Actinomyces* sp S6 Spd3, and *Atopobium parvulum*. The same gut bacteriome communities were observed at 6 months for the PACS group, indicating that *B. wexlerae* and *B. longum* have an inverse correlation with PACS after six months, showing that these beneficial microbes could have a putative role in recovery from COVID-19. After 6 months, the bacteriome picture of the patients who did not develop PACS was similar to the non-COVID-19 controls, while at the same time point, the patients who developed PACS had a distinct and less diverse bacteriome in comparison to the aforementioned groups. At six months, patients with PACS showed significantly lower levels of *Bacteroides vulgatus*, *Blautia obeum, Collinsella aerofaciens, Faecalibacterium prausnitzii,* and *Ruminococcus gnavus* in comparison to the non-COVID-19 controls. Patients with PACS had reduced levels of other symbionts including species of *Roseburia* and *Faecalibacterium*. These two genera have known immunomodulatory effects and promote immune homeostasis by producing short chain fatty acids. These fatty acids have been linked to various roles including modification of phagocytosis, chemotaxis, cell-proliferation, and anti-inflammatory effects [177]. Liu et al. also observed that the gut bacteriome in patients with PACS showed increased abundances of the urea cycle pathway, the L-citrulline biosynthetic pathway and the L-ornithine biosynthetic Ⅱ pathway [176].

Box 2Post-acute COVID-19 syndrome (PACS).PACS can be diagnosed for patients who suffer from COVID-19 symptoms even after 4-weeks of the initial onset of clinicals signs. PACS can cause non-specific symptoms including fatigue, low-grade fever, cough, dyspnea, loss of smell or taste, myalgia, and depression [178]. Other organ systems can be affected during PACS, including the cardiovascular, pulmonary, renal, and nervous systems. Persistent symptoms associated with post-acute COVID-19 syndrome seem to impact physical and cognitive function, health-related quality of life, and participation in society [179]. Patients with prior health conditions may be more vulnerable to PACS [180,181]. However, data are still lacking concerning the effects of this syndrome on the various organ systems in the human body.In a study by Davis et al., 2021, the common symptoms in patients after 6 months were fatigue, post-exertional malaise, and cognitive dysfunction [182]. Patients with long COVID report prolonged, multisystem involvement and significant disability. By seven months, many patients have not yet recovered and continue to experience various symptoms.

Another recent study that provided further information about the interplay between the GB, the immune system and SARS-CoV-2 was by Xu et al. [183]. The authors used metagenomic shotgun sequencing and immune profiling in their study that included COVID-19 patients (mild and severe) and healthy controls. They found the GB of severe patients to be less diverse as compared to the other two groups, with increased representation of *Akkermansia muciniphila*, *Bacteroides cellulosilyticus*, *Bacteroides ovatus, Enterococcus avium, Enterococcus durans,* and *Enterococcus faecium*. The genera that decreased in numbers in the severe group compared to mild and healthy groups included *Coprococcus*, *Dialister, Klebsiella,* and *Roseburia.* The mild COVID-19 group had higher abundances of *Mediterraneibacter*, *Blautia*, *Streptococcus*, *Anaerostipes, Anaerobutyricum,* and *Ruminococcus*. In their immunological analyses, they made connections with the immune profiles and the abundances of bacterial species. They observed that higher abundances of *Blautia obeum, Coprococcus catus*, *Coprococcus comes*, and *Roseburia intestinalis* had a positive correlation with T-cell counts, indicating that these bacteria could play potential roles in the immune response to SARS-CoV-2. They also observed various enzyme markers that were present in higher amounts due to increases in specific bacterial spp. For instance, they found a positive correlation between the creatine kinase isoenzyme (marker for cardiovascular impairment) and two spp., *Bacteroides cellulosilyticus* and *Akkermansia muciniphila*. A similar correlation was observed between these spp. and aspartate transaminase (marker for liver injury). Conclusively, by using pathway enrichment analysis of the GB, the authors observed numerous metabolic pathways that were differentially enriched between the mild and severe COVID-19 groups. For example, the enrichment of the super pathway of polyamine biosynthesis II was decreased in the severe group compared to the mild group. A review of the literature suggested that polyamines have an active role in immunity. They are required for normal T-cell proliferation and can also inhibit the production of pro-inflammatory cytokines (IL-1β and IL-6) in mouse macrophages [184,185]. 

As the GB is the most densely populated anatomical site in the human body with trillions of bacterial cells, the effects of antibacterial treatments during COVID-19 infections could be most profound here [186]. Another important aspect of gut dysbiosis is the translocation of pathogenic bacterial spp. from the gut via the bloodstream into other body locations. This could lead to serious bacterial secondary infections [187]. In a comparative study, Venzon and co-workers compared GB dysbiosis in a mouse model experimentally infected with SARS-CoV-2 with stool samples of COVID-19 patients, and also analyzed whether a dysbiotic gut bacteriome could result in translocation of bacterial spp. to other anatomical locations [188]. In the mouse model, a change in bacterial diversity was observed with increased abundance of the genus *Akkermansia* and reduced populations of the genera *Clostridium* and *Erysipelothrix.* The authors also checked whether the disruption in the gut bacterial communities could cause the leaky gut phenomenon. In the infected mice signs of gut barrier dysfunction were observed. The plasma concentration of Fluorescein 121 isothiocyanate-dextran (administered by oral gavage) was not significant and similar non-significant results were observed for markers (citrulline, fatty acid-binding protein, and lipopolysaccharide-binding protein) that predict permeability of the gut barrier. However, the group of mice that underwent drastic diversity changes in the gut bacterial populations showed higher concentrations of Fluorescein 121 isothiocyanate-dextran. In addition, extremely high populations of the genus *Akkermansia* were observed. The changes in key epithelial cells, and signs of gut barrier compromise suggested diminution of bacteriome diversity in the mice. In the human study, COVID-19 patients had high numbers of spp. belonging to the phyla Firmicutes, Bacteroidetes, and Proteobacteria. In 21 patients, a positive correlation was observed between reduced bacterial diversity and bacterial secondary infections, indicating that reduced diversity in the gut bacteriome could lead to secondary infections. This analysis revealed that the genus *Faecalibacterium* was negatively associated with bloodstream infections. This bacterium is in an immune-supportive Clostridiales genus and is a member of the healthy gut bacteriome. Decreased populations could lead to disruption of the gut barrier. Based on our current understanding, the important players of the GB with their dominant metabolic pathways, and changes in cytokine levels in the gut are shown in Figure 9.

## 10. Metabolites of the Human Bacteriome Versus COVID-19

Recent interest has emerged regarding about the role of metabolites produced by bacterial residents of the human body, their potential as direct anti-viral agents against SARS-CoV-2, and their involvement as drivers of immune-modulation that may help lower the inflammatory effects of the virus [189,190]. Past literature about the actions of bacterial metabolites against viral infections have been reviewed [191]. A recent study by Piscotta and colleagues identified three bacterial metabolites, isopentenyl adenosine (an adenosine analogue), tryptamine (an indolamine metabolite), and 2,5-bis(3-indolylmethyl) pyrazine to be potent inhibitors of SARS-CoV-2 [192]. All three metabolites are related structurally and functionally to the synthetic Food and Drug administration approved drugs (Remedisvir, Fluvoxamine, and Favipiravir, respectively) that have been evaluated in COVID-19 clinical trials. The three metabolites were observed to be produced by four commensals of the human bacteriome including *Bacteroides caccae*, *Prevotella nigrescens* (both produce Isopentyl adenosine), *Ruminococcus gnavus* (tryptamine), and *Micrococcus luteus* (2,5-bis(3-indolylmethyl)pyrazine). Interestingly, the first of these bacterial spp. are commensals of the human gut [193], while the last species is a commensal of the human skin but has pathogenic potential [194]. As the aforementioned metabolites had not previously been associated with the human bacteriome, data are lacking that could determine if they ever accumulate to physiologically relevant concentrations. Additionally, any predicted ecological roles of these metabolites in a complex host, bacteriome, and any potential virus tripartite interaction is still unknown. However, the similarity between commensal antiviral metabolites and the Food and Drug administration approved antivirals is intriguing and should be subject for future studies on these metabolites.

A group of metabolites that could be of paramount importance with respect to the COVID-19 war are short chain fatty acids (SCFAs). These are produced by members of the human bacteriome and include three compounds: acetate, butyrate, and propionate [195]. The mechanisms of action and the potential roles in immunomodulation of these SCFAs have been reviewed extensively [196,197,198]. A study by Kim et al. showed that SCFAs produced by bacterial residents of the human gut can enhance gene expression and promote activation of B-cells, thus regulating antibody responses to pathogens [199]. As these metabolites affect B-cell mediated immune responses in the gut and systemic tissues, a potential anti-COVID-19 action may be mediated by these metabolites in the human gut. Another study by Zhang et al. found that SCFA levels remained impaired even after resolution of the disease in a group of COVID-19 patients [200], thus indicating that SARS-CoV-2 could directly influence the metabolic products encoded by bacterial genomes in the human body. 

The SCFAs can exert anti-inflammatory effects by activation of anti-inflammatory immune cells and inhibition of the signaling pathways for inflammation [201]. Butyrate in particular has been shown to exert an anti-inflammatory effect [202], and in COVID-19, butyrate may reduce hyperinflammation. Additionally, higher numbers of butyrate-producing bacteria in the gut helped to reduce respiratory viral infections in patients who underwent a kidney transplant [203]. In addition, butyrate can reduce gut hyperpermeability by increasing the expression of tight-junction proteins. This helps decrease endotoxemia and inflammation associated with leaky gut syndrome [204]. Butyrate also prevents hyperinflammation through modulation of the functions of M2 macrophages and regulatory T cells. It also inhibits infiltration by neutrophils, and by extension, upregulates arginase 1 downregulates TNF, nitric oxide synthase-2, IL-6, and IL-12b [205]. Recently, it has been shown that butyrate protects the host from viral infection by down-regulation of genes that have been proven to be essential for SARS-CoV-2 infection, such as ACE2, and up-regulating TLR antiviral pathways in gut epithelial organoids models [206]. In the human body, butyrate is metabolized rapidly, leading to its low bioavailability [207]. To counter this, specific dosages of oral butyrate could be used in COVID-19 patients to achieve anti-inflammatory effects while avoiding any unwanted effects on the immune response. 

*Bacteroides* spp. are key players in immunomodulation and could be key producers of metabolites that harness immunomodulation. *Bacteroides fragilis* synthesizes capsular polysaccharides such as polysaccharide A, which has proven positive effects on the immune system [208]. This polysaccharide is packaged into outer membrane vesicles and delivered to host cells. Other metabolites from various gut bacterial genera include vitamins, amino acid derivatives and glycolipids [209]. These bacterial metabolites could be used in tandem with other SARS-CoV-2 prophylaxes and therapies to achieve better clinical outcomes for patients. As of now, limited data are available about studies on the anti-COVID-19 effects of bacterial metabolites. The need of the hour is to bio-informatically predict the immune-modulatory and anti-viral potential of these metabolites, harness these compounds, and check their virucidal activity in both in vitro and in vivo setups. 

In this section of the review, we have focused only on metabolites that are direct by-products of resident bacteria. However, metabolites produced by other microbes such as fungi, algae, etc., could also be tested for potential anti-SARS-CoV-2 activity, and also for positive effects on the immune system that could help lower viral infection and dissemination. 

## 11. Concluding Remarks

The COVID-19 pandemic is not the first viral pandemic to hit humankind and will certainly not be the last. The current pandemic, however, has reminded us how novel pathogenic viruses can emerge and evolve at a rapid rate to cause havoc to the health of humans and other animals.

The genomes of SARS-CoV-2 have been subject to genetic variations by mutations followed by selection and recombination [210]. It has been shown that point mutations provide the major driving force behind the evolution of the virus. These mutations have been reported in genes encoding the spike protein, especially the receptor binding domain, and the N-terminal domain [211]. Studies have shown that SARS-CoV-2 has an estimated mutation rate of 1.12 × 10^−3^ mutations per site-year, which is similar to the reported rates of other coronaviruses (SARS-CoV-1, and MERS-CoV) [212]. The evolution of the virus results in challenges for diagnostic purposes (quantitative real time polymerase chain reaction), as primer designing for amplification of genes of the virus becomes a major problem. In one study, Osorio and Navez checked the efficacy of the oligonucleotides/primers used by the World Health organization for detection of the major SARS-CoV-2 genomes [213]. The authors observed that 79% of the oligonucleotide/primer binding sites were mutated in at least one genome. The findings were not regarded as sequencing errors, as the results were consistent over different testing sites, thus the observed variants were regarded as “true variants”. The authors concluded that the present primers/oligonucleotides are ineffective in detecting 14% of the SARS-CoV-2 variants due to genetic variations in the genome of the virus. The evolution of the virus can also influence the efficacy of vaccination. The World Health Organization utilizes a naming system for SARS-CoV-2 variants by using Greek alphabets. Four of the variants of concern include Alpha, Beta, Delta and Gamma (for details, see the review of Duong, 2021) [214]. The genetic variations in the Alpha variants have a limited effect on the efficacy of COVID-19 vaccines. However, some of the commonly used vaccines have partial protection against the Beta and Delta variants. This could be due to the high levels of neutralizing antibodies elicited and the robust and broad nature of the T-cell response elicited by several of the vaccines. Data from various booster vaccination programs indicate that mRNA vaccine boosters could provide protection against the omicron variant [215]. 

The time is right to understand how our bacteriomes (oral, nasal, lung, and gut) can help us not only to limit virus spread, but also to limit the loss of human life. In having millions of bacterial cells, we heavily depend on our bacterial residents in the fight against all sorts of foreign pathogens (viruses, bacteria, protozoa, and fungi), and the current scenario with SARS-CoV-2 is no different. Beneficial bacterial spp. as well as their immunomodulatory and anti-viral metabolites may help the human host in the battle against SARS-CoV-2. The third major player in the COVID-19 battle, “the human immune system”, needs to be in a homeostatic state to counter SARS-CoV-2. This homeostasis seems to be heavily influenced by the human bacteriome. 

Currently, numerous vaccines are in use for prophylaxis of COVID-19 [216]. Another point of interest, which is currently overlooked, could be the potential role that the human bacteriome could have on vaccine efficacy. Through this communication, we have already summarized the relationship between the human bacteriome and the immune system, with context to how the former could modulate the latter in a negative or positive manner. Achieving desired immune responses to vaccines is a complex process that is controlled by numerous variables such as health and nutritional status, intrinsic host factors (sex, age, and genetic makeup), and environmental factors (geographical location). As the human bacteriome substantially influences the immune system, it can be safely assumed that the resident bacteria of the human body will influence vaccine responses. In the near future, having an estimate of bacterial diversity could be a good option prior to vaccine administration. Additionally, future studies could be set up to check vaccine efficacy in participants on the basis of their different bacteriomes. However, as described by Lynn et al., the path to establishing a clear connection with specific bacterial taxa and vaccine efficacy is not straightforward [217]. However, murine studies could be of importance in identifying novel probiotic strains that assist in increasing vaccine immunogenicity. As the variables that affect vaccine effectiveness could also in turn affect the bacteriome composition, it is important to use novel multi-omics and vaccinology approaches to come to a consensus on how to analyze the variables and vaccine efficacy. For this purpose, approaches similar to those of Hagan et al. [218], who analyzed the correlation between gut bacterial load and influenza vaccine efficacy, could be used. 

A burning issue involves the identification of those bacterial genera whose presence and/or increased populations can limit viral activities. Through this communication, we have tried to identify beneficial and/or probiotic bacterial spp. that could be used in the formulation of potential probiotic therapeutics for COVID-19 patients. However, why certain bacterial spp. seem to act as superior probiotics as compared to other species during the course of COVID-19 infection is still a question that requires attention. Additionally, the proteome of these species is of interest, as the proteins and metabolites encoded by these species contribute to their probiotic characteristics. Further studies on their proteomes would be necessary to advance our understanding of their probiotic potential. In addition, the pathogens that thrive during COVID-19 infection should be subject to future bioinformatic and clinical studies. Bacterial secondary infections during COVID-19 are a major player in disease severity and patient mortality. The COVID-19 storm may have lost its momentum for the time being, but, as with any adversary, we should be ready for its next move (mutation, etc.) and improving the beneficial potential of our bacterial dwellers could be the way forward. 

## Figures and Tables

**Figure 1 cells-12-01213-f001:**
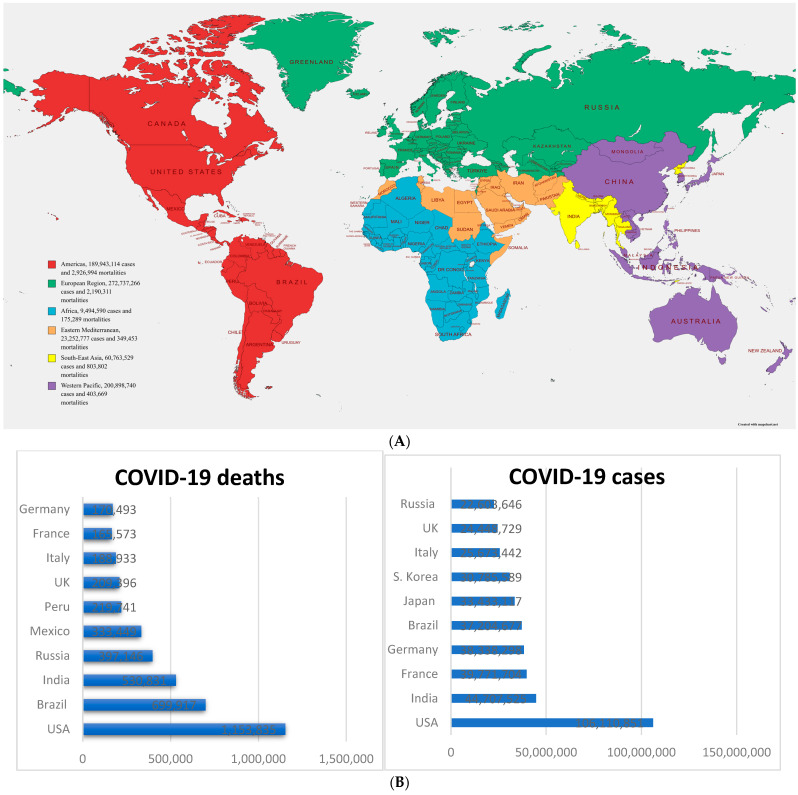
(**A**) The data have been adapted from https://www.who.int/publications/m/item/weekly-epidemiological-update-on-covid-19, accessed on 28 February 2023 REF [2]. The world map shows the current situation (February 2023) of the COVID-19 pandemic in terms of cases and mortalities across different WHO regions. Europe has the most cumulative cases, with the numbers currently standing at 272,737,266, and the most mortalities (2,926,994) have been reported in the Americas. (**B**) The total number of COVID-19 cases and deaths in the ten most-affected countries. The data are from https://www.worldometers.info/coronavirus, accessed on 22 March 2023 REF [3].

**Figure 2 cells-12-01213-f002:**
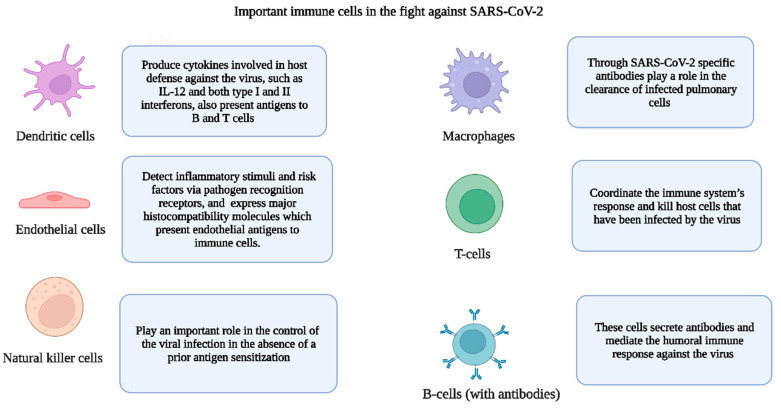
Brief descriptions of the important immune cells in the fight against COVID-19 are shown. The information used in the figure has been adapted from [30,31,32,33].

**Figure 3 cells-12-01213-f003:**
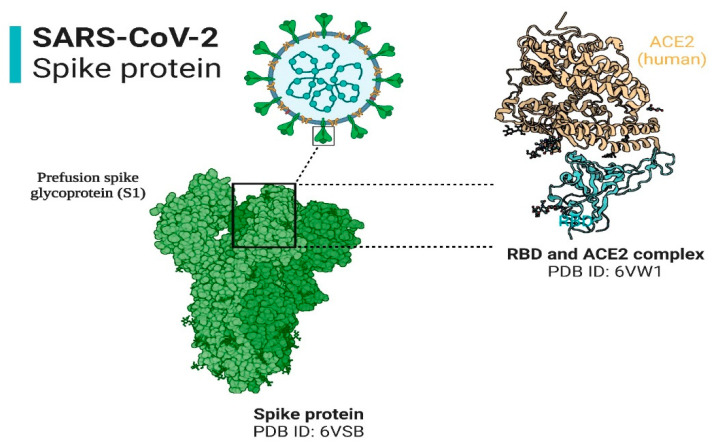
Structure of SARS-CoV-2 and the spike protein (**left**), and interaction of the receptor binding domain (RBD) with ACE2 receptor (**right**).

**Figure 4 cells-12-01213-f004:**
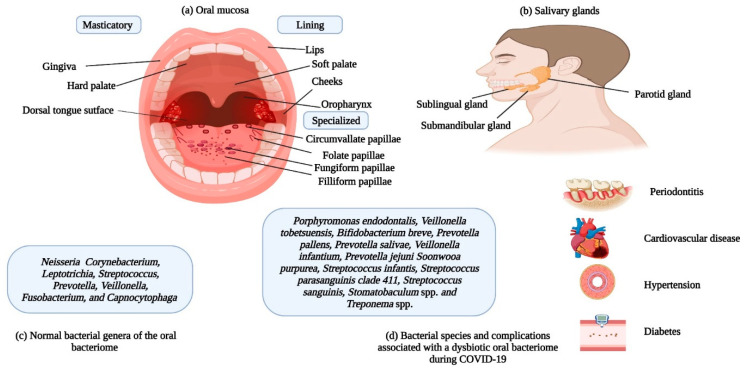
(**a**) The three parts of the oral mucosa are shown. (**b**) The three salivary glands. (**c**) Names of the common bacterial genera in the human oral cavity. (**d**) Names of bacterial spp. that thrive in a dysbiotic oral bacteriome during COVID-19 infection are shown; also, the associated health complications/disease conditions are revealed. The data used in the figure have been adapted from Refs [44,45,46].

**Figure 5 cells-12-01213-f005:**
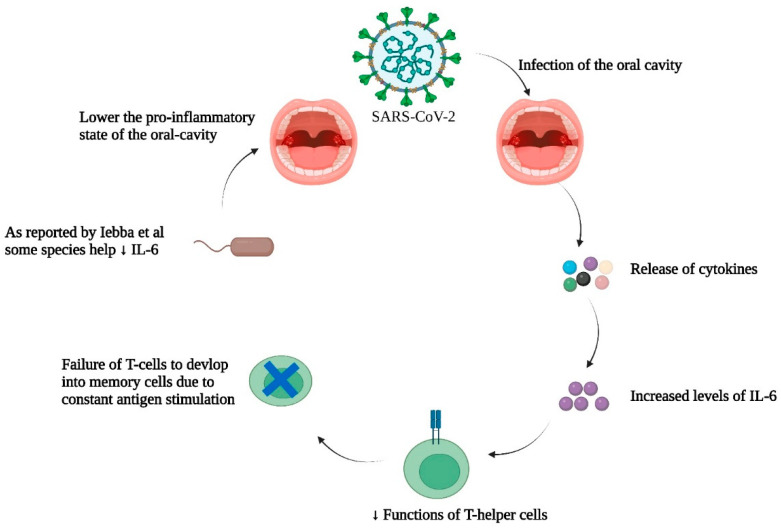
SARS-CoV-2 multiplication in the oral cavity causes the release of cytokines, particularly higher levels of IL-6. This decreases the function of T-helper cells, and T-cells fail to develop into T-memory cells for future encounters with the same antigen. It has been reported by Iebba et al. that some members of the oral bacteriome help to lower IL-6 levels in the oral cavity, thus resulting in a lower pro-inflammatory state that would help to avoid constant antigen stimulation and counter the cytokine storm. Data have been adapted from references [73,74].

**Figure 6 cells-12-01213-f006:**
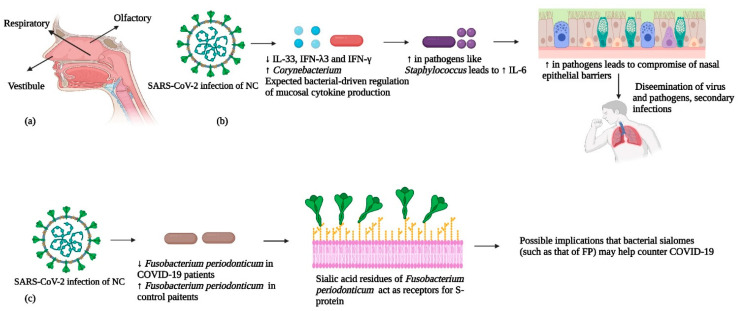
(**a**) The three parts (respiratory, olfactory, and vestibule) of the nasal cavity are shown. (**b**) During COVID-19 infection, major bacterial and immunological changes take place in the nasal cavity including increases in the genera *Corynebacterium* and *Staphylococcus* and changes in the levels of different cytokines. This could lead to disruption of epithelial barriers and cause secondary bacterial infections in other body locations. (**c**) Sialic acid residues of the bacterium *Fusobacterium periodonticum* may help to counter COVID-19 infections. Data used in the figure are from references [84,85,86].

**Figure 7 cells-12-01213-f007:**
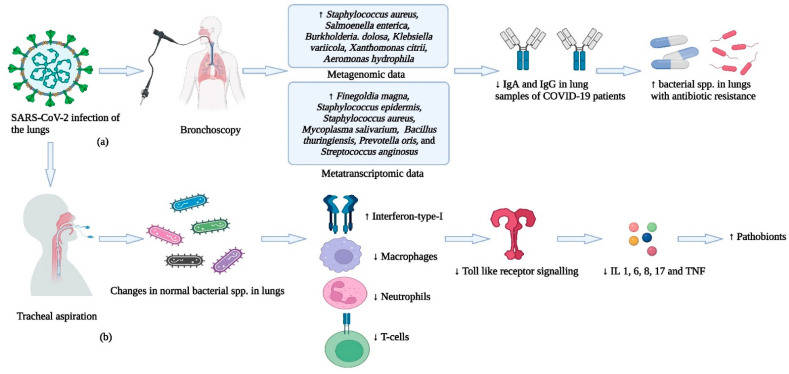
(**a**). The Metagenomic and Metatranscriptomic data from Sulaiman et al. show an increase in various bacterial spp. in lung samples of COVID-19 patients (samples were collected by bronchoscopy). In the COVID-19 patients, there are increases in IgA, IgG, and antibiotic resistant bacterial spp. (**b**) In another study (Tsitsiklis et al.), lung samples were collected by tracheal aspiration. There were changes in the normal bacterial flora of the lungs, changes in immune cells and cytokines, and increased populations of pathobionts. Data used for the figure were taken from references [136,137].

**Figure 8 cells-12-01213-f008:**
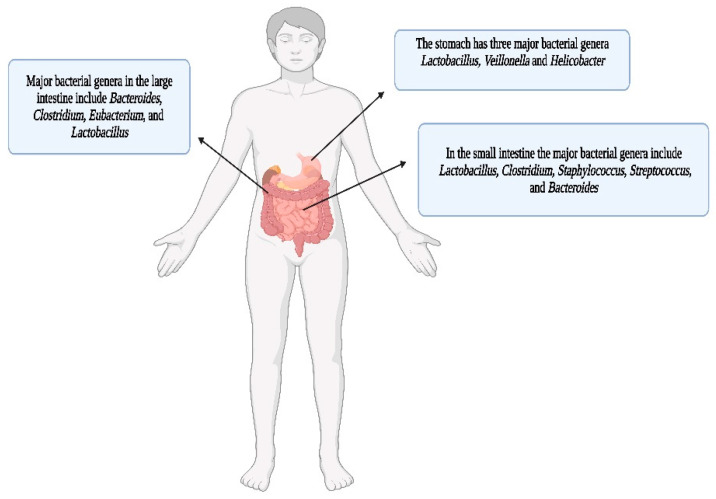
Parts of the human gut with dominant bacterial genera.

**Figure 9 cells-12-01213-f009:**
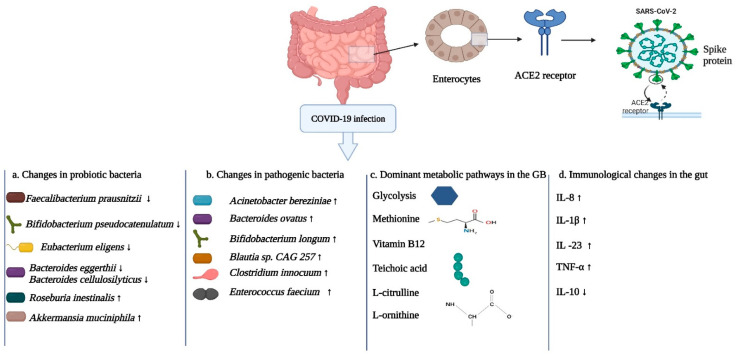
Enterocytes in the intestine have ACE2 receptors to which SARS-CoV-2 can bind. During COVID-19 infection, there are various changes in the bacterial populations (probiotic and pathogenic) of the intestine. Additionally, the bacterial spp. that thrive in the gut during COVID-19 infection have preferences towards certain metabolic pathways, which enable their survival and multiplication. These pathways could also assist in viral infection. Various changes in the cytokine levels in the gut also take place. Data adapted from references [183,189].

## Data Availability

Not applicable.

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
