# Peer review of "Understanding the Relationship of the Human Bacteriome with COVID-19 Severity and Recovery"

_cells, 2023, doi:10.3390/cells12091213_

Round 1

Reviewer 1 Report

The author explains how the bacteriome of different anatomical sites (oral, nasal, lung and gut) in a patient with COVID-19 infection was modified, in comparison with the bacterial population (bacteriome) of healthy humans, where the immune system plays a main role.

This manuscript is original, I could not find any other similar manuscript in the databases. We know that the microbiome composition contributes to maintaining the homeostasis of the body and how its alteration may lead to a dysbiosis that may cause disease, and in the case of a viral infection, it may lead to a more severe complication.

The conclusions are consistent with the evidence and arguments presented and they address the main question posed.

Minor revision:

Line 604. However, the densities of microbial populations differ not only in various parts of the gut (stomach) this organ is no part of gut.

Line 608 Veilonella, this word must be write correctly throughout of text and Figure 8 (Veillonella); the same manner intestine  in the Figure 8.

Line 657. It is recommends that the authors to review the manuscript carefully and correct the name of the microorganisms, punctuation marks.

Line 666. CAG257 no italics, According no capital letter.

Line 704.  and.

Author Response

Reviewer 1 comments

  1. Line 604. However, the densities of microbial populations differ not only in various parts of the gut (e.g., the stomach; this organ is not part of gut).

            We have rectified the mistake as pointed out by the reviewer in the revised version.

  1. Line 608 Veilonella, this word must be write correctly throughout of text and Figure 8 (Veillonella); the same manner intestine  in the Figure 8.

We thank the reviewer for pointing out this typo, we have now corrected the mistake as advised.

  1. Line 657. It is recommends that the authors to review the manuscript carefully and correct the name of the microorganisms, punctuation marks.

In light of the comment, we have carefully reviewed the revised version of the manuscript. We believe the previous mistakes (wrong names of bacterial species/strains) and punctuation errors have been rectified.

  1. Line 666. CAG257 no italics, According no capital letter, Line 704.  and.

As suggested the changes have been made in the revised manuscript.

We would like to thank Reviewer 1 for the positive comments about the manuscript and the important suggestion to improve it.

Reviewer 2 Report

The manuscript titled "Understanding the Relationship of the Human Bacteriome with COVID-19 Severity and Recovery" presents an interesting perspective on the potential role of bacterial species in the human bacteriome in influencing COVID-19 severity and recovery. However, certain aspects of the manuscript need further improvement.

Major points:

1. The statement claiming that the human body has more bacterial cells than its own and, therefore, making us more bacterial than humans is highly questionable and should be removed. Instead, the authors should consider recent research (https://doi.org/10.1371/journal.pbio.1002533) indicating that the ratio of human to bacterial cells is likely to be 1:1.

2. The authors should provide more context on the current state of the pandemic and better explain the concept and implications of COVID-19 becoming endemic in its interaction with the human bacteriome. For example, the statement "At the time of this communication, the Coronavirus disease (COVID-19) may be on way to an endemic form" should be better explained in the manuscript.

3. The authors should acknowledge that the classification of immune cells is a complex and evolving field and that identifying specific cell types as "important" risks overlooking the crucial contributions of other immune cells. Instead, the authors should refer to a possible selection of immune cells they consider to be most relevant to the discussion at hand and explain their criteria. For instance, the statement in Figure 2 regarding "the important immune cells" is potentially misleading and oversimplifies the complexity of the immune system.

4. The authors should provide more information on the implications of SARS-CoV-2 evolution in the context of the manuscript. For example, the sentence "evolve at a rapid rate to cause havoc to the health of humans and other animals" is too general and does not provide specific information on the consequences of SARS-CoV-2 evolution. The authors should consider citing recent research (https://doi.org/10.1016/S1473-3099(20)30435-7, https://doi.org/10.1128/mbio.02979-21) to highlight objective categories, such as diagnostic and vaccination efficacy, affected by SARS-CoV-2 evolution.

Minor points:

1. Figure 1 needs improvement. The world map is entirely green, making it difficult to interpret the distribution of COVID-19 cases worldwide. It would be helpful to paint the continents with a gradient color scale to better portray the number of cases and/or mortality. Furthermore, the two bar plots are too large and could be decreased in size and arranged side-by-side in a panel to make the figure easier to interpret.

2. The manuscript requires copyediting in some parts. The authors should carefully proofread the manuscript to correct occasional grammatical errors and awkward phrasings.

Overall, this manuscript has potential, but the authors should carefully address the points above to further raise its quality to publication standards.

Author Response

Reviewer 2 comments

Major points

  1. The statement claiming that the human body has more bacterial cells than its own and, therefore, making us more bacterial than humans is highly questionable and should be removed. Instead, the authors should consider recent research (https://doi.org/10.1371/journal.pbio.1002533) indicating that the ratio of human to bacterial cells is likely to be 1:1.

We thank the reviewer for this valuable suggestion, as we ourselves believed that bacterial cells are much greater in numbers in comparison to human cells in the body.  As suggested, we have removed the statements throughout the manuscript that mentioned “human cells were outnumbered by bacterial cells in the body”. We have also added the valuable information that was provided in the suggested paper and cited it. 

  1. The authors should provide more context on the current state of the pandemic and better explain the concept and implications of COVID-19 becoming endemic in its interaction with the human bacteriome. For example, the statement "At the time of this communication, the Coronavirus disease (COVID-19) may be on its way to an endemic form" should be better explained in the manuscript.

We have now provided up-to-date information in light of recent research on how COVID-19 has taken an endemic form. However, despite our through search of the relevant literature, we have been unable to find any research/information that implicates the human bacteriome in any effect on the endemicity of SARS-CoV-2.

  1. The authors should acknowledge that the classification of immune cells is a complex and evolving field and that identifying specific cell types as "important" risks overlooking the crucial contributions of other immune cells. Instead, the authors should refer to a possible selection of immune cells they consider to be most relevant to the discussion at hand and explain their criteria. For instance, the statement in Figure 2 regarding "the important immune cells" is potentially misleading and oversimplifies the complexity of the immune system.

As suggested by the reviewer, we have redone Figure 2 to highlight the immune cells that have been shown to be major players during COVID-19 infection. We have also cited relevant research that proposes the mentioned functions of the cells during infection. We firmly believe the Figure 2 makes more sense with respect to the topic in question, as the previous figure was a mere overview of important cells. We thank the reviewer for the valuable suggestion.

  1. The authors should provide more information on the implications of SARS-CoV-2 evolution in the context of the manuscript. For example, the sentence "evolve at a rapid rate to cause havoc to the health of humans and other animals" is too general and does not provide specific information on the consequences of SARS-CoV-2 evolution. The authors should consider citing recent research (https://doi.org/10.1016/S1473-3099(20)30435-7, https://doi.org/10.1128/mbio.02979-21) to highlight objective categories, such as diagnostic and vaccination efficacy, affected by SARS-CoV-2 evolution.

In light of this comment and the two papers we have added the appropriate information has been included in the Conclusion section, page , lines. [ADD THIS INFORMATION]

Minor points:

  1. Figure 1 needs improvement. The world map is entirely green, making it difficult to interpret the distribution of COVID-19 cases worldwide. It would be helpful to paint the continents with a gradient color scale to better portray the number of cases and/or mortality. Furthermore, the two bar plots are too large and could be decreased in size and arranged side-by-side in a panel to make the figure easier to interpret.

We have now improved Figure 1 and added different colors according to each WHO region. Also, we have decreased the sizes of the bar plots and arranged them side-by-side as suggested by the reviewer. We do believe that Figure 1, now gives a better visual interpretation of COVID-19 cases and mortalities.

  1. The manuscript requires copyediting in some parts. The authors should carefully proofread the manuscript to correct occasional grammatical errors and awkward phrasings. Overall, this manuscript has potential, but the authors should carefully address the points above to further raise its quality to publication standards.

In light of the reviewer’s suggestion, we have carefully proofread the revised version and tried to address any grammatical errors and awkward phrasings. We hope the modifications will help to increase the scientific rigor of the manuscript.

Round 2

Reviewer 2 Report

I'm pleased to see that the authors have taken reviewer comments into account and have made significant improvements to the manuscript.